# Impact of terminal cleaning in rooms previously occupied by patients with healthcare-associated infections

Marc Verhougstraete[1]*, Emily Cooksey[1], Jennifer-Pearce Walker[1], Amanda M. Wilson[1], Madeline S. Lewis[2], Aaron Yoder[1], Gabriela Elizondo-Craig[1], Munthir Almoslem[1], Emily Forysiak[1], Mark H. Weir[1,2,3]

1 Mel and Enid Zuckerman College of Public Health, The University of Arizona, Tucson, Arizona United States of America, 2 Sustainability Institute, The Ohio State University, Columbus, Ohio, United States of America, 3 College of Public Health, The Ohio State University, Columbus, Ohio, United States of America

* mverhougstraete@email.arizona.edu

**Data Availability Statement:** All data are available on GitHub: https://github.com/mverhougstraete/HAI-data.git.

## Abstract

Healthcare associated infections (HAIs) are costly but preventable. A limited understanding of the effects of environmental cleaning on the riskiest HAI associated pathogens is a current challenge in HAI prevention. This project aimed to quantify the effects of terminal hospital cleaning practices on HAI pathogens via environmental sampling in three hospitals located throughout the United States. Surfaces were swabbed from 36 occupied patient rooms with a laboratory-confirmed, hospital- or community-acquired infection of at least one of the four pathogens of interest (i.e., *Acinetobacter baumannii* (*A. baumannii*), methicillin resistant *Staphylococcus aureus* (MRSA), vancomycin resistant *Enterococcus faecalis/faecium* (VRE), and *Clostridioides difficile* (*C. difficile*)). Six nonporous, high touch surfaces (i.e., chair handrail, bed handrail, nurse call button, desk surface, bathroom counter near the sink, and a grab bar near the toilet) were sampled in each room for Adenosine Triphosphate (ATP) and the four pathogens of interest before and after terminal cleaning. The four pathogens of interest were detected on surfaces before and after terminal cleaning, but their levels were generally reduced. Overall, *C. difficile* was confirmed on the desk (n = 2), while MRSA (n = 24) and VRE (n = 25) were confirmed on all surface types before terminal cleaning. After cleaning, only MRSA (n = 6) on bed handrail, chair handrail, and nurse call button and VRE (n = 5) on bathroom sink, bed handrail, nurse call button, toilet grab bar, and *C. difficile* (n = 1) were confirmed. At 2 of the 3 hospitals, pathogens were generally reduced by >99% during terminal cleaning. One hospital showed that VRE increased after terminal cleaning, MRSA was reduced by 73% on the nurse call button, and VRE was reduced by only 50% on the bathroom sink. ATP detections did not correlate with any pathogen concentration. This study highlights the importance of terminal cleaning and indicates room for improvement in cleaning practices to reduce surface contamination throughout hospital rooms.

**Funding:** The US Centers for Disease Control and Prevention (CDC) provided support towards the project design, protocol development, and technical review this manuscript (75D30118C02916). The CDC did not have a role in the data collection or analysis.

**Competing interests:** The authors have declared that no competing interests exist.

## Introduction

Healthcare associated infections (HAIs) pose risks to hospitalized patients, with 4.5% of U.S. hospitalized patients obtaining an HAI during their stay [1], resulting in annual excess costs of $30 billion to the healthcare sector [2]. Along with direct outcomes of HAI occurrence, an indirect consequence is the continued reliance on antibiotic to treat HAIs. Among HAI pathogens, antibiotic resistance is observed in *Acinetobacter baumannii* (*A. baumannii*), methicillin resistant *Staphylococcus aureus* (MRSA), vancomycin resistant *Enterococcus faecalis/faecium* (VRE), and *Clostridioides difficile* (*C. difficile.*) [3–6]. Direct measurements of these pathogens can be costly and timely, and thus proxies for general cleanliness are often used, such as adenosine triphosphate (ATP). ATP is a biochemical indicator of biological material. Despite ATP being characteristic of bacterial metabolism and regularly utilized as a cleanliness tool, its measurements have been shown to be poorly correlated with pathogens in healthcare settings [7], making it a questionable proxy for environmental cleanliness and HAI risk.

Hand hygiene and surface cleaning are currently the primary interventions to prevent HAIs. Hand hygiene education efforts initially increase staff awareness and compliance, but this effect decreases overtime, reducing its effectiveness as an HAI mitigation strategy and putting overreliance on one intervention [8, 9]. Surface cleaning research that addresses real-world efficacy rates, compliance rates, and predicted impacts on patient health is lacking [10]. Despite limited data quantifying the impacts of surface cleaning efficacies and compliance rates on HAIs, it is well acknowledged that surfaces have the potential to harbor pathogens which contribute to indirect transmission [11, 12]. Specifically, high touch surfaces (e.g., bed handrails and nurse call remotes) and soft surfaces (e.g., sheets and curtains) have been shown to play a significant role in pathogen transmission in healthcare settings [13–17]. Additionally, there is evidence of increased risk of contracting a HAI from exposure to the associated pathogen if the previous room occupant was infected with the same pathogen [18, 19].

Terminal cleaning protocols have varying levels of real-world efficacy in reducing HAI causing pathogens on surfaces and subsequent HAI cases [20]. Using hypochlorite solutions, *C. difficile* infection (CDI) rates were reduced to <3.7 per 1000 cases [21, 22] and bioburden levels on surfaces were reduced by 50% [23]. MRSA bioburden levels were decreased below detectable levels on 98.8% of surfaces using hydrogen peroxide vapor, but terminal cleaning with detergent-based (containing 5–15% non-ionic surfactant and 5–15% cationic surfactant) only decreased MRSA below detectable levels on 44% of surfaces [24]. VRE is challenging to remove by cleaning, even with bleach solutions in hospital settings [25, 26]. Many HAI and surface cleaning studies focus on one or two pathogens rather than a suite of HAI causative agents. Assessing HAI pathogen prevalence and survivability in healthcare environments allows for a better understanding of the role that surface cleaning has in the reduction of pathogen exposures across multiple types of HAI associated pathogens.

HAIs continue to plague hospitals despite increased hand and environmental hygiene efforts which act as single transmission stop points [9]. To investigate underlying causes of this persistent public health problem, opportunities for infection via multiple pathogens and infection control strategies for addressing multiple pathogens need to be examined. Therefore, this study involved the quantification of surface concentrations for four HAI associated pathogens before and after terminal cleaning. The objectives were to 1. Quantify *C. difficile*, MRSA, *A. baumannii*, and VRE on surfaces of hospital rooms previously occupied by a confirmed infected patient before and after terminal cleaning using culture-based methods, 2) confirm culture results using mass spectrometry, and 3) compare confirmed results to ATP measurements.

## Scope of research

This research was designed to assess the overall effectiveness of terminal cleaning (e.g., final room cleaning when a patient vacates and before a new patient occupies the room) on the retention of four HAI causing bacteria in patient rooms. An additional objective of determining the efficacy of using ATP to validate the cleaning of rooms was included. The study was conducted at three hospitals in AZ, OH, and GA. In environmental health science practice, the lower the retention (higher reduction) of bacteria in the room, the lower the risk to the next occupant. Thus, this is an initial study critical to improving our understanding of exposure pathways and eventually health risks associated with remaining HAI pathogens.

## Methods

### Sample site selection

Surface samples were collected from three total hospitals located in the Southwest (Arizona) (hospital 1), Midwest (Ohio) (hospital 2), and Southeast (Georgia) (hospital 3) regions of the United States. Patient rooms were selected for surface sampling if the patient being discharged had a laboratory-confirmed infection with at least one of the study target pathogens. Environmental sampling pre-cleaning was performed after the patient was discharged but prior to terminal cleaning. Environmental sampling post-cleaning was performed immediately after terminal cleaning and before the next patient occupied the room. Researchers were not provided any information on the time since daily cleaning prior to discharge or on the patient, beside the identified pathogen. No patient data was recorded.

Six nonporous, high touch surfaces were sampled in each room before and after terminal cleaning and informed by touch frequencies measured in hospitals [27–29]. The six surfaces (i.e., chair handrail, bed handrail, nurse call button, desk surface, bathroom counter near the sink, and a grab bar near the toilet) were preselected for sampling in all hospital rooms. Descriptions of the surfaces are presented in Table 1.

Terminal cleaning protocols for each of the three hospitals were compared. The protocol for hospital 1 indicated use of unspecified germicidal wipes for bathroom and surfaces throughout patient room and instructed bleach specific wipes be used for *C. difficile* positive patient rooms. The protocol for hospital 2 indicated the use of Clorox bleach wipes and Versasure wipes (quaternary ammonium chloride based, Clorox® Healthcare®) for all surfaces in all rooms. The protocol for hospital 3 indicated use of unspecified germicidal wipes for bathroom and surfaces throughout patient room and no additional measures for high-risk scenarios.

### Surface sampling

Surfaces were sampled for microorganisms using sterile cellulose sponge sticks pre-wetted with 10 mL neutralizing buffer (Fisher Scientific, NC1536284). As a result of shipping delays

**Table 1. Description of surface samples.**

| Surface | Description | Surface area dimensions (cm) | Sampled area (cm²) |
|---|---|---|---|
| Bathroom counter sink | Sink located in patient bathroom | 35.6 x 50.8 | 100 |
| Bed handrail | Rail attached to patient bed used for patient support | 99.1 x 4.45 x 5.08 | 100 |
| Chair handrail | Chair located in patient room | 53.3 x 5.08 | 100 |
| Desk | Patient desk located near bed | 38.1 x 85.1 | 100 |
| Nurse call button | Nurse call remote located at patient bed | 7.62 x 20.3 | 113 |
| Toilet grab bar | Bar located near toilet for patient support | 20.3 x 5.08 x 3.81 | 100 |

of material, a small subset of samples (n = 14) were collected using cellulose sponge sticks pre-wetted with 10 mL Letheen broth and were processed immediately after collection. All surfaces were assayed for ATP (SystemSURE Plus, Hygiena LLC, Camarillo, CA, USA) before and after terminal cleaning in an area (100 cm$^2$) adjacent to microorganism sampling area. Based on previous research to maximize recovery potential from the surface sampling, each surface was sampled for microorganisms and ATP over 100 cm$^2$ for all items except the nurse call button [30]. The nurse call button was divided into 113 cm$^2$ quadrants to simplify sampling due to asymmetrical shape. For surfaces smaller than 100 cm$^2$, the surface area was measured and divided in half, then the results were converted to microorganisms per 100 cm$^2$. Environmental parameters (temperature and relative humidity) were recorded for each room at the time of sampling. A negative control was performed for each sampling event. This was completed by opening a swab, breaking the sponge end off into the bag, and leaving the sponge in the bag during room sampling. The bag was then closed and processed in the laboratory alongside the surface samples.

Samples were stored at 4˚C and processed within 24 hours of collection. The plastic stick was removed from each sample and 45 mL of sterile 1x (phosphate buffered saline + 0.02% polysorbate 80) PBST were added to each sample. Sponges were agitated for 1 minute at 200 RPM in a stomacher (Seward stomacher 400 circulator, West Sussex, United Kingdom). Sponges were aseptically squeezed to remove remaining liquid, and sponges were discarded. Liquid was transferred to sterile 50 mL conical centrifuge tubes and centrifuged for 20 minutes at 2700 x g. Supernatant was pipetted out to a final volume of 5–7.5 mL, the tubes were vortexed for 2 minutes to dislodge the pellet into the remaining supernatant, and the final volume measured and recorded.

## Microbial analysis

Each sample was quantitatively assayed for presence of *A. baumannii*, MRSA, VRE, and *C. difficile*. Sample eluent was spread plated in duplicate (500 μL per plate) onto the appropriate selective media for each organism. CHROMagar™ *Acinetobacter* media (DRG International Inc.) was used for isolation of *Acinetobacter* spp. with red colonies counted as positive colonies. HardyCHROM™ MRSA agar (Hardy Diagnostics) was used to isolate methicillin-resistant *S. aureus* with deep pink to magenta colonies counted as positive. CHROMagar™ VRE (DRG International Inc.) was used to isolate vancomycin resistant *E. faecalis/faecium* with pink to mauve colonies counted as positive. Samples plated onto CHROMagar™ Acinetobacter, HardyCHROM™ MRSA, and CHROMagar™ VRE were incubated at 37± 2˚C for 24–48 hours. Cycloserine Cefoxitin Fructose Agar with Horse Blood and Taurocholate (CCFA-HT, Anaerobe Systems) was used for isolation of *C. difficile* with large, grey, slightly filamentous, and low umbonate to flat colonies counted as positive. Samples plated onto CCFA-HT media were incubated at 37 ± 2˚C for 48–72 hours under anaerobic conditions created using anaerobic chambers and multiuse sachets (BD Diagnostics). The number of colonies that grew on each selective media was used to define the presumptive concentration for each organism and surface. All culture results were presented as CFU/area. Presumptive results are presented to highlight the need for confirmation testing when performing HAI associated pathogen assessment with selective media.

Confirmation testing was completed for *C. difficile*, MRSA, or VRE. For any *C. difficile*, MRSA, or VRE growth on respective selective media, three representative colonies were chosen and frozen in a 1 to 4 tryptic soy broth to glycerol freezing media and stored at -80˚C. If CFU/sample was less than three, as many colonies as available were frozen and stored. The genus and species of frozen colonies were confirmed using matrix assisted laser desorption/

ionization–time of flight (MALDI-TOF) mass spectrometry (Brukar MALDI Biotyper). This testing confirmed visual assessments of colonies from selective media and identification of pathogens. Confirmation testing was not completed for *Acinetobacter* spp. Non-*Acinetobacter* organisms (*e.g. Pseudomonas* and *Sphingomonas*) may grow on the CHROMagar™ *Acinetobacter* media. However, such occurrences would be most expected from wet surfaces such as around sinks or from biofilm samples. From the dry hospital surfaces targeted in this study, non-*Acinetobacter* growth on the selective media was not expected and therefore additional confirmation of isolates was not performed. A sample was considered a confirmed detection when 1) a sample had at least one presumptive CFU on selective agar and 2) at least one presumptive CFU was confirmed as the target pathogen through MALDI-TOF. The final confirmed concentration for each organism was calculated using the presumptive concentration multiplied by the fraction of isolates confirmed positive through MALDI-TOF assessment with results presented as CFU/swabbed area.

## Statistical analysis

Statistical analysis was completed using R version 3.5.1 (r-project.org). Descriptive statistics on pathogen concentrations and percent positive were calculated for each organism by surface and location. An assessment of the mean microbial reductions across all surfaces and hospitals was performed to identify significant organismal reductions following terminal cleaning. Due to the small, confirmed sample size, bacterial concentrations across surfaces and hospitals was not performed.

## Results

### Environmental sampling

During this project, 36 sampling events occurred across three hospitals with each event involving the swabbing of six surfaces before and after terminal cleaning ($N_{surface\ swabs}$ = 432 samples), and each sample was processed for four microorganisms ($N_{total\ assays}$ = 1,728). Across all samples, the highest pre-cleaning concentrations (presumptive and confirmed) were all from MRSA (chair handrail, nurse call button, toilet grab bar). After terminal cleaning, the highest presumptive concentrations were from MRSA on the nurse call button. After terminal cleaning, the bed handrail and nurse call button both had measurable MRSA and VRE concentrations. The presumptive and confirmed metadata for all hospitals, pathogens, and surfaces is provided in Table 2. The major surfaces of concern across all hospitals were the desk surface and the nurse call button where 21 rooms had measurable confirmed concentrations of at least one organism pre-cleaning. Post-cleaning, confirmed organisms were detected on the nurse call button (n = 4), the toilet grab bar (n = 3), chair handrail (n = 2), bed handrail (n = 2), and the bathroom counter sink (n = 1). No organisms were detected post-cleaning on the desk surfaces. VRE was detected on the toilet grab bar in two rooms and the bed handrail in one room which did not have detectable VRE pre-cleaning. When the post-cleaning, confirmed concentration of all three hospitals were investigated, no significant differences by surface or by location were identified. Although there were no significant differences, mean concentrations for the nurse call button and toilet grab bar were highest of all surfaces and mean MRSA and VRE concentration were highest of all organisms.

At the individual hospital level, hospital 1 had the highest pre-cleaning concentrations for *C. difficile* on toilet grab bars, MRSA on chair handrails, and MRSA on nurse call button. After cleaning, the highest confirmed concentrations included MRSA on chair handrails, MRSA on nurse call button, and VRE on bed handrail. At hospital 2, the highest presumptive concentrations were observed pre-cleaning on the bed handrail (*C. difficile*) and post-cleaning on the

**Table 2. Presumptive and confirmed geometric mean concentrations (CFU 100 cm$^{-2}$) for the four targeted organisms on surfaces of all hospitals before and after terminal cleaning.**

| Organism | Location | Pre-cleaning surface concentrations (CFU 100 cm$^{-2}$) | | Post-cleaning surface concentrations (CFU 100 cm$^{-2}$) | |
| --- | --- | --- | --- | --- | --- |
| | | Presumptive | Confirmed | Presumptive | Confirmed |
| *Acinetobacter* | Bathroom counter sink | 0.35 | NC | <0.08 | NC |
| | Bed handrail | <0.08 | | <0.08 | |
| | Chair handrail | <0.08 | | <0.08 | |
| | Desk | <0.08 | | <0.08 | |
| | Nurse call button | <0.07 | | 0.00 | |
| | Toilet grab bar | <0.08 | | <0.08 | |
| *C. difficile* | Bathroom counter sink | <0.08 | <0.08 | <0.08 | <0.08 |
| | Bed handrail | 0.11 | <0.08 | <0.08 | <0.08 |
| | Chair handrail | 0.14 | <0.08 | <0.08 | <0.08 |
| | Desk | <0.08 | <0.08 | <0.08 | <0.08 |
| | Nurse call button | <0.07 | <0.08 | <0.07 | <0.07 |
| | Toilet grab bar | 0.24 | <0.08 | <0.08 | <0.08 |
| MRSA | Bathroom counter sink | 0.67 | 0.19 | <0.08 | <0.08 |
| | Bed handrail | 0.35 | <0.08 | 0.09 | <0.08 |
| | Chair handrail | 1.75 | 1.41 | <0.08 | <0.08 |
| | Desk | 0.69 | 0.41 | 0.14 | <0.08 |
| | Nurse call button | 1.36 | 1.20 | 1.21 | 0.28 |
| | Toilet grab bar | 1.07 | 0.09 | 0.09 | <0.08 |
| VRE | Bathroom counter sink | <0.08 | 0.09 | <0.08 | <0.08 |
| | Bed handrail | <0.08 | <0.08 | <0.08 | <0.08 |
| | Chair handrail | 0.08 | <0.08 | <0.08 | <0.08 |
| | Desk | 0.09 | <0.08 | <0.08 | <0.08 |
| | Nurse call button | <0.08 | <0.08 | <0.07 | <0.07 |
| | Toilet grab bar | 0.12 | <0.08 | 0.10 | 0.15 |

NC: *Acinetobacter* isolates not confirmed for this project; Less than values indicate concentrations were below the limit of detection.

chair handrail (*C. difficile*). However, following MALDI tests, *C. difficile* was not confirmed on any surface, while VRE and MRSA were confirmed on the nurse call button (VRE: pre-cleaning, MRSA: post-cleaning). At hospital 3, the highest presumptive pre-cleaning concentrations were associated with MRSA on the desk and toilet grab bar while the highest presumptive post-cleaning concentrations were associated with MRSA on the nurse call button. The highest confirmed pre-cleaning pathogens (MRSA and VRE only) were found on bathroom counter sink, desk, and toilet grab bar. After terminal cleaning, only MRSA was confirmed on the toilet grab bar.

## Detections

Data from all hospitals included a total of 51 out of 149 confirmed detections pre-cleaning and 12 out of 65 confirmed detections post-cleaning. MRSA, VRE, and *C. difficile* were presumptively detected on every surface type pre-cleaning. However, across all hospitals, MRSA, VRE, and *C. difficile* were ultimately only confirmed on 11.3%, 11.8%, and 1% of pre-cleaning samples, respectively. After cleaning, *C. difficile*, MRSA, and VRE were confirmed on 2.8%, 2.4%, and 0.47% of post-cleaning samples, respectively, and were found on all surfaces except the desk. The nurse call button had four combined confirmed detections post-cleaning across all

**Table 3. Presumptive and confirmed detections and frequency for pre- and post-cleaning at all hospitals.** A sample was considered a confirmed detection when a sample had at least one presumptive CFU on selective agar and at least one presumptive CFU was confirmed as the target pathogen through MALDI-TOF.

| | N | Pre-cleaning | | | | Post-cleaning | | | |
|---|---|---|---|---|---|---|---|---|---|
| | | Presumptive detections | Confirmed detections | Presumptive detection percentage (%)* | Confirmed detection percentage (%)* | Presumptive detections | Confirmed detections | Presumptive detection percentage (%)* | Confirmed detection percentage (%)* |
| **Pathogens** | | | | | | | | | |
| *Acinetobacter* | 212 | 12 | 0 | 5.66 | 0.00 | 7 | 0 | 3.30 | 0.00 |
| *C. difficile* | | 39 | 2 | 18.4 | 0.94 | 20 | 1 | 9.43 | 0.47 |
| MRSA | | 61 | 24 | 28.8 | 11.3 | 27 | 6 | 12.7 | 2.83 |
| VRE | | 37 | 25 | 17.5 | 11.8 | 11 | 5 | 5.19 | 2.36 |
| **TOTAL** | | **149** | **51** | | | **65** | **12** | | |
| **Surfaces** | | | | | | | | | |
| Chair handrail | 144 | 16 | 4 | 11.1 | 2.78 | 12 | 2 | 8.33 | 1.39 |
| Bed handrail | | 22 | 7 | 15.3 | 4.86 | 15 | 2 | 10.4 | 1.39 |
| Nurse call button | | 25 | 10 | 17.4 | 6.94 | 10 | 4 | 6.94 | 2.78 |
| Desk surface | | 23 | 11 | 16.0 | 7.64 | 8 | 0 | 5.56 | 0.00 |
| Bathroom sink counter | | 33 | 9 | 22.2 | 6.25 | 9 | 1 | 6.25 | 0.694 |
| Toilet grab bar | | 30 | 10 | 20.8 | 6.94 | 11 | 3 | 7.64 | 2.08 |
| **TOTAL** | | **149** | **51** | | | **65** | **12** | | |

*Detection percentage calculated for

• **Pathogen specific surface** detection percentage calculated as: number of samples with CFU (only specific organism and all surfaces) / total pathogen specific samples (6 surfaces*36 rooms)—Pre-cleaning n = 212; post-cleaning n = 207; *e.g. Acinetobacter* = 12/212 = 5.66%;

• **Surface specific pathogen** detection percentage calculated as: number of samples with CFU (only specific surface and all 4 organisms) / total surface specific samples 144 (4 organisms * 36 room); *e.g.* chair handrail = 16/144 = 11.1%

hospitals. A summary of all presumptive and confirmed detections across all hospitals and surfaces is detailed in Table 3.

At hospital 1, VRE was the most frequently detected and confirmed pathogen during post-cleaning surface sampling with the nurse call button having the highest confirmed detection frequency post-cleaning. At hospital 2, tests confirmed one target organism pre-cleaning (VRE on nurse call button) and one target organism post-cleaning (MRSA on the nurse call button). At hospital 3, MRSA on the toilet grab was the only confirmed organism detected post-cleaning.

There was a significant correlation (p = 0.008) between ATP and presumptive VRE pre-cleaning concentrations at hospital 1. There were no significant relationships between ATP and other organisms during pre- or post-cleaning at hospital 1($\rho > 0.23$). This is highlighted in Fig 1 which shows the ATP measurements along with all presumptive organism detection concentrations pre- and post-cleaning. ATP measures on surfaces were routinely >0 relative light units (RLUs) in the three hospitals before and after cleaning.

Mean microbial reductions across all surfaces were statistically significant (p<0.05) after post-cleaning for *C. difficile* and MRSA at hospital 1, VRE at hospital 2, and *C. difficile* and VRE at hospital 3. There were statistically significant (p<0.05) differences in concentrations before and after cleaning in hospital 1 on the bathroom sink (all organisms) and desk surface (all organisms) and in hospital 3 on the toilet grab bar (VRE), nurse call button (*C. difficile*), and bathroom sink (all organisms). Hospital 1 saw notable reductions in confirmed *Acinetobacter*, MRSA, and *C. difficile*, but VRE increased post-cleaning. This average increase in VRE

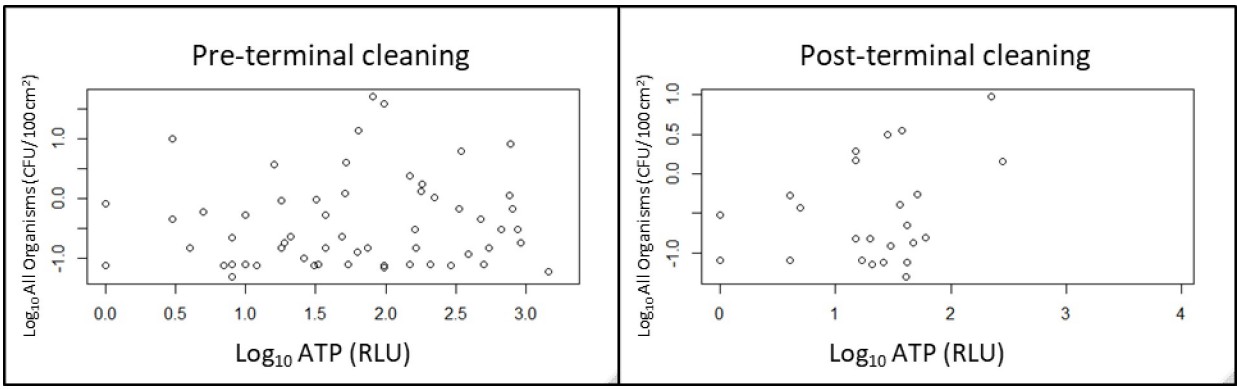

**Fig 1. Scatter plots of ATP versus all microorganism on all surfaces at hospital 1 pre- and post-terminal cleaning.**

was driven by the nurse call button which had higher concentrations post cleaning. Excluding the nurse call button and toilet grab bar, VRE concentrations post-cleaning were 50% lower than pre-cleaning, on average. At hospital 2, VRE was the only organism detected and confirmed (nurse call button) pre-cleaning and was reduced below detectable levels (described as 100% reduction) via terminal cleaning. The lowest percent reductions at hospital 3 were observed for confirmed MRSA on the toilet grab bar (87.5% reduction). A summary of average percent reductions for each organism, surface, and hospital are provided in Table 4.

## Discussion

The environmental sampling of hospital rooms previously occupied by patients with a confirmed MRSA, *C. difficile*, VRE, or *A. baumannii* infection were targeted to quantify the efficacy of terminal hospital cleaning practices on pathogen occurrence. Hospital policy prevented the research team from obtaining data on which target pathogen corresponded with the sampled room, or whether the rooms were ICU or non-ICU rooms. Terminal cleaning protocols differed slightly between the three tested hospitals with two of the three noting practices that would be effective at killing *C. difficile* and one had no mention of specialty practices for any pathogen. Specific information on cleaning practices such as methods of cleaning, cleaning products used, cleaning compliance rates, and the order of surfaces cleaned could not be obtained due to hospital policies. Across all hospitals, terminal cleaning practices generally reduced, but did not completely remove (below detectable limits), surface pathogens and, in some instances, may have introduced pathogens to areas where they were previously absent. However, on an individual hospital level, the number of confirmed organisms above our limit of detection was not large enough to determine the effectiveness of a single terminal cleaning protocol. One limitation of this study is that the cleaning effect of physically wiping in the correct manner in comparison to use of disinfectants could not be assessed. Additional sampling efforts are required to test individual terminal cleaning protocol effects on specific pathogens and surfaces.

Overall, microorganism detections and their concentrations decreased by surface from pre- to post-cleaning. During one sampling event in one room at hospital 1, the confirmed concentration of VRE increased from pre-cleaning (below limit of detection) to post-cleaning (0.15 CFU cm$^{-2}$) sampling. This suggests cross contamination in cleaning hospital rooms is possible and requires further research on the spread of HAIs between fomites during cleaning practices. Surface contamination and transfer of microorganisms between fomites is complicated. While no information was gathered on patient behaviors, EVS cleaning practices and

**Table 4. Average percent reductions for each confirmed organism and surfaces at the three hospitals when organisms were detected before cleaning.** Blank cells indicate no detection.

| | Organism | Hospital 1 | Hospital 2 | Hospital 3 |
|---|---|---|---|---|
| OVERALL | *Acinetobacter* genus | | - | |
| | *C. difficile* | 100% | - | - |
| | MRSA | 94.5% | - | 98.9% |
| | VRE | 2.78% | 100.0% | 100% |
| Bathroom sink | *Acinetobacter* genus | - | - | - |
| | *C. difficile* | - | - | - |
| | MRSA | 100% | - | 100% |
| | VRE | 50.0% | - | 100% |
| Bed handrail | *Acinetobacter* genus | - | - | - |
| | *C. difficile* | - | - | - |
| | MRSA | 100% | - | - |
| | VRE | 100% | - | - |
| Chair handrail | *Acinetobacter* genus | - | - | - |
| | *C. difficile* | - | - | - |
| | MRSA | 99.1% | - | - |
| | VRE | 100% | - | 100% |
| Desk surface | *Acinetobacter* genus | - | - | - |
| | *C. difficile* | 100.0% | - | - |
| | MRSA | 100.0% | - | 100.0% |
| | VRE | 100.0% | - | 100.0% |
| Nurse call button | *Acinetobacter* genus | - | - | - |
| | *C. difficile* | - | - | - |
| | MRSA | 73.43% | - | 100.0% |
| | VRE | -287.5%* | 100.00% | - |
| Toilet grab bar | *Acinetobacter* genus | - | - | - |
| | *C. difficile* | - | - | - |
| | MRSA | 100.0% | - | 87.5% |
| | VRE | 100.0% | - | 100.0% |

NA: No organisms detected before cleaning

* Negative percent reduction is indicative of either potential growth or cross contamination during terminal cleaning.

behaviors, frequency and duration of environmental cleaning, frequency of healthcare worker visit to patient room, airflow within hospital rooms, overall patient colonization rates in each hospital, infection prevalence rates at time of admission in each hospital, or average length of stay, these factors warrant future assessment as they could contribute to the transfer of pathogens between fomites. Future studies should also examine the role of cloth material and detergent wipes [31] and the surface roughness [32, 33] in this transfer continuum. Increases in concentration following terminal cleaning could also indicate disaggregation during cleaning, increasing detectable CFUs. Given each of the three hospital cleaning protocols call for germicidal agents, the environmental sampling suggests additional cleaning efforts or prescribed surface cleaning orders are required to reduce pathogen occurrence and/or cross-contamination.

The desk was the only surface post-cleaning where no organisms were confirmed whereas the nurse call button and the toilet grab bar exhibited the highest post-cleaning confirmed rates of any surfaces. This may suggest surface accessibility plays a role in cleaning effectiveness. That is, flat and open surfaces (e.g., desk surface) are easier to clean than surfaces with small topographic intricacies, electronic interfaces (e.g., nurse call button), or surfaces in

difficult to reach spaces (e.g., toilet grab bar). Additionally, discrepancies in cleaning may result in surfaces not being sufficiently cleaned, as previously indicated by high variability in surface cleaning efficacy within a hospital setting [34]. In this study all three hospitals demonstrated that no statistically significant differences were seen in confirmed concentrations between pre-cleaning and post-cleaning, regardless of location. This is likely a result of the low confirmed sample size for all pathogens. Further studies assessing the method of cleaning and role of cleaning staff would provide necessary information to better understand differences in surface cleanliness.

ATP results from the current study were not a reliable indicator of pathogen presence or cleaning efficacy. Across all hospitals, ATPs were measured on nearly all surfaces before and after cleaning. This supports prior hospital studies in the USA [35, 36], and Brazil [37] that found no significant relationship of ATP and pathogens. While numerous studies have explored the association between ATP levels and the presence of pathogens in hospital settings, the existing body of literature reveals a spectrum of findings. Some studies report poor relationships [38–41], while others indicate significant associations [42–47] between ATP measurements and pathogen presence. This variability underscores the complexity of assessing cleaning practices and the importance of utilizing multi assessment approaches to understand environmental hygiene. This approach is crucial for refining and enhancing infection control strategies, ultimately contributing to the overall improvement of patient safety in healthcare facilities.

Cleaning interventions are implemented to address broad spectrum HAIs or to target specific pathogens [48]. A broad approach creates challenges in identifying the most effective intervention for a hospital's specific need while a pathogen specific approach may not be protective enough. Further, overlap of interventions does not necessarily result in additive risk reduction, rather there is the potential for interference and inefficiencies. These concerns warrant further mechanistic risk and epidemiologically driven transmission modeling tools to explore and optimize intervention options within a central framework. The data from this current study can help fill such gaps and enhance decision-making capabilities.

## Conclusion

This study demonstrates the potential for terminal hospital cleaning practices to reduce pathogens on high touch surfaces. Eradication of pathogens below detectable levels was not always attained, which is what the preponderance of terminal cleaning protocols and products are designed, developed, and advertised to achieve. ATP measurement did not indicate pathogen presence and should therefore not be used as the only measure of hospital surface safety. The three partner hospitals had terminal cleaning protocols but only one of them specifically addressed *C. difficile*. The variability in cleaning efficacy and potential cross-contamination detected in this study indicates that current terminal cleaning protocols should be amended and standardized to protect patients from being exposed to pathogens left on surfaces by previous patients. Protocols should follow current CDC guidance [49] and should be enhanced to more effectively target MRSA and VRE on surfaces, as these were the most commonly identified microorganisms on surfaces after terminal cleaning. Given the uncertainties related to terminal cleaning's effectiveness, movement to a risk-based framework in which terminal cleaning efforts can be translated to anticipated HAI outcomes is needed.

## Acknowledgments

The authors gratefully acknowledge the generous efforts of Laura Rose and Sujan Reddy who supported project design, protocol development, and technical reviews of the manuscript. The

authors acknowledge and appreciate the environmental sampling and processing efforts from Dr. Lisa Casanova at Georgia State University.

## Author Contributions

**Conceptualization:** Marc Verhougstraete, Mark H. Weir.

**Data curation:** Marc Verhougstraete, Jennifer-Pearce Walker, Amanda M. Wilson, Mark H. Weir.

**Formal analysis:** Marc Verhougstraete, Jennifer-Pearce Walker, Mark H. Weir.

**Funding acquisition:** Marc Verhougstraete, Mark H. Weir.

**Investigation:** Marc Verhougstraete, Emily Cooksey, Jennifer-Pearce Walker, Amanda M. Wilson, Madeline S. Lewis, Mark H. Weir.

**Methodology:** Marc Verhougstraete, Jennifer-Pearce Walker, Amanda M. Wilson, Mark H. Weir.

**Project administration:** Marc Verhougstraete, Jennifer-Pearce Walker, Mark H. Weir.

**Resources:** Marc Verhougstraete.

**Supervision:** Marc Verhougstraete, Mark H. Weir.

**Visualization:** Marc Verhougstraete, Emily Cooksey, Aaron Yoder.

**Writing – original draft:** Marc Verhougstraete, Emily Cooksey, Jennifer-Pearce Walker, Amanda M. Wilson, Aaron Yoder, Gabriela Elizondo-Craig, Mark H. Weir.

**Writing – review & editing:** Marc Verhougstraete, Amanda M. Wilson, Gabriela Elizondo-Craig, Munthir Almoslem, Emily Forysiak, Mark H. Weir.

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
