## [Decision Letter · Decision Letter 0]

11 Jan 2024

PONE-D-23-27473Impact of terminal cleaning in rooms previously occupied by patients with healthcare-associated infectionsPLOS ONE

Dear Dr. Verhougstraete,

Thank you for submitting your manuscript to PLOS ONE. After careful consideration, we feel that it has merit but does not fully meet PLOS ONE’s publication criteria as it currently stands. Therefore, we invite you to submit a revised version of the manuscript that addresses the points raised during the review process.

We look forward to receiving your revised manuscript.

Kind regards,

Eili Y. Klein, PhD

Academic Editor

PLOS ONE

“The authors gratefully acknowledge the generous funding for this research from the US Centers for Disease Control and Prevention (CDC) under Contract number: 75D30118C02916. Specifically, we acknowledge Laura Rose and Sujan Reddy who provided support towards the project design, protocol development, and technical reviews of reports and manuscript. The authors acknowledge and appreciate the environmental sampling and processing efforts from Dr. Lisa Casanova at Georgia State University.”

“MV, MW

US Centers for Disease Control and Prevention (CDC) under Contract number: 75D30118C02916

https://www.cdc.gov/index.htm

Yes: CDC scientists helped define appropriate confirmation tests for each pathogen”

“No”

6. Please remove your figures from within your manuscript file, leaving only the individual TIFF/EPS image files, uploaded separately. These will be automatically included in the reviewers’ PDF.

7. Please keep your tables as part of your main manuscript and remove the individual files. Please note that supplementary tables (should remain/ be uploaded) as separate "supporting information" files

Additional Editor Comments:

Sorry for the delay in review. The original reviewer went out on parental leave and there was difficulty securing another reviewer. I agree with the reviewers that the paper is useful and adds to the literature on room cleaning, but that it needs some work. Some of the issues are methodological (e.g., why no negative control) some were questions on the statistical choices, and some were just comments on the readability. A major issue was a description of the surfaces. It appears this may have been in a citation, but that citation shows up as an error. Perhaps add some details about the surfaces here as well as correcting the citation. The other issue noted was the concern about increases at certain hospitals either in whole or part. There was confusion about what increased and how significant was that increases. There was particular concern around the tables which are not always clear.

Reviewers' comments:

Reviewer's Responses to Questions

**Comments to the Author**

1. Is the manuscript technically sound, and do the data support the conclusions?

Reviewer #1: Partly

Reviewer #2: Yes

2. Has the statistical analysis been performed appropriately and rigorously? 

Reviewer #1: Yes

Reviewer #2: Yes

3. Have the authors made all data underlying the findings in their manuscript fully available?

Reviewer #1: No

Reviewer #2: Yes

4. Is the manuscript presented in an intelligible fashion and written in standard English?

Reviewer #1: Yes

Reviewer #2: Yes

5. Review Comments to the Author

Reviewer #1: Summary

This study highlights the impact of terminal room cleaning on removing pathogens from high-touch surfaces. This is an absolutely important subject, particularly with potential implications for the role of environment in transmission of HAIs.

The study includes the collection and analysis of environmental samples from 36 patient rooms at 3 major hospitals located in 3 different US states. The findings suggest that while complete eradication of pathogens to undetectable levels, which is the goal of most terminal cleaning protocols, was not consistently achieved, concentration levels of Acinetobacter baumannii, MRSA, VRE, and C diff. were moderately to significantly reduced after the cleaning procedure across different surface types. One interesting finding was the increase in VRE concentration on nurse call buttons after cleaning in one of the hospitals, which the authors argued to be potentially due to cross contamination. Another interesting finding, which agrees with other studies, was that using only ATP measurement was shown to be insufficient for assessing hospital surface cleanliness. The authors have provided suggestions about amending and standardizing current terminal cleaning protocols to protect patients from exposure to residual pathogens on surfaces left by previous patients, with a particular focus on MRSA and VRE as the most frequently found pathogens on hospital surfaces after terminal cleaning. In light of uncertainties regarding the effectiveness of terminal cleaning, the study suggests transitioning to a risk-based framework that can link terminal cleaning protocols to expected HAI outcomes.

I found the study interesting and suitable for publication in PLOS ONE, though I have a few questions that I would like to be addressed before publication.

Major comments/questions

1. The authors have provided a brief discussion of the potential impacts of the differences in cleaning protocols and compliance across the three sites on their findings, in addition to other limitation of their study (which is shared with other similar studies). I think it would help to make a more comprehensive discussion if other factors that may contribute to variabilities and rare but significant discrepancies (such as VRE increase in one of the hospitals) were also incorporated, factors such as differences in cleaning staff diligence, patient colonization rates, admission prevalence rate, average length of stay, frequency of HCW visits, and other factors that affect pathogen shedding and transfer and may differ across rooms, wards, and hospitals.

2. Discussion, Paragraph 2: “on at least one occasion, detections and/or concentrations increased from pre-cleaning to post-cleaning sampling. This occurred for C. difficile on chair handrail, MRSA on bed handrail…” While the “one” occasion was covered in the results section for VRE on nurse call buttons, other instances of such observations, where pathogen concentration was increased after terminal cleaning, was not explained neither could be interpreted from the results (Tables 2 and 3), for example C. difficile on chair handrail. I am not sure if these other instances were isolated cases in one or just a few rooms, or if the findings were not reliable due to below-detectable concentrations. What could be the reason? Please elaborate.

3. The one occasion where VRE concentration was increased after terminal cleaning is interesting. While the authors have provided their interpretation of the potential causes, it would make it easier for readers to objectively interpret such a rare but impactful case if more information was provided, for example, was this observed for all nurse call buttons (all rooms), if not how many of the rooms, and whether there could be anything different about those specific rooms or their occupying patients. This deserves a little deeper investigation.

4. Statistical analysis: what method was used for multiple hypothesis tests (detections) to correct for family-wise type I errors? If such a consideration was not necessary, please justify by explaining in the paper the scale of detection tests (i.e., whether detection tests were done on a surface-, room-, or hospital-level) in relation to the comparison scale which is surface type at each hospital.

Minor comments

1. Abstract: Please spell out what ATP stands for, the first time it is used.

2. Abstract: Incomplete sentence: “Surfaces from 36 occupied … and Clostridioides difficile (C. difficile)).”

3. Abstract: Missing comma before “desk surface” in the following statement: “Six nonporous, high touch surfaces (i.e., chair handrail, bed handrail, nurse call button desk surface, bathroom counter near the sink, and a grab bar near the toilet) ...”

4. Introduction, Paragraph 1: “ATP as a biochemical indicator of biological material.” Incomplete sentence. Please complete or merge with the previous or next sentence.

5. Methods, Sample Site Selection, Paragraph 2: Missing comma before “desk surface” in the following statement: “The six surfaces (i.e., chair handrail, bed handrail, nurse call button desk surface, bathroom counter near the sink, and a grab bar near the toilet) ...”

6. Methods, Sample Site Selection, Paragraph 2: Please address the cross-referencing error.

7. Methods, Sample Site Selection, Paragraph 2: “If a pre-selected item was not present …” An example or two for the replacements could help clarify, also how often this was the case?

8. Methods, Sample Site Selection, Table 1: To improve readability, please align cells, preferably to the top left corner of the cells, or use borders.

9. Methods, Detections, below Figure 1: “At hospital 2, VRE was the only organism detected and confirmed (n nurse call button) pre-cleaning…” Please correct the typo inside the parentheses.

10. Discussion, Paragraph 4: “Other studies, however, have shown poor relationship relationships [38–41] …” Please remove the extra “relationship.”

Reviewer #2: Thank you for this informative paper in this important area of infection prevention and control in the hospital setting. I have some comments that revolve primarily around making the paper clearer, and easier for the reader to understand and contextualize. It includes using a denominator to give context to the numbers you are referrring to.

-The abstract needs to include the number of rooms being evaluated for each in order to make sense of these results.

-The statement "ATP detections did not correlate with any pathogen concentration and thus are not recommended for assessing cleanliness or safety." is too conclusive based on the data shown. ATP measures organic matter, and would be better correlated with overall aerobic colony counts, it is not specific to multidrug resistant organisms.

-In methods, there is an error: "Descriptions of the surfaces are presented in Error!Reference source not found.."

-Include use of negative control in the micro methods section if used, or if not used explain why.

- Presumptive and confirmed concentrations needs to be explained in the methods. From my reading/understanding only AB isolates from wet surfaces were confirmed as these may be non-baumanii. But then it seems to apply to all and is very unclear. Table 2 is also very unclear.

- Detections needs a denominator to put the “51” and “12” on the first line in context. The rest of the section is unclear, and does not seem to align with the methods.

-Discussion: needs to be clearer - Other studies, however, have shown poor relationship relationships [38–41] or significant

relationships [42–47] between ATP and pathogen presence.

6. PLOS authors have the option to publish the peer review history of their article (what does this mean?). If published, this will include your full peer review and any attached files.

Reviewer #1: **Yes: **Fardad Haghpanah

Reviewer #2: No

---

## [Author Response · Author response to Decision Letter 0]

7 Mar 2024

Reviewer #1: Summary

This study highlights the impact of terminal room cleaning on removing pathogens from high-touch surfaces. This is an absolutely important subject, particularly with potential implications for the role of environment in transmission of HAIs. The study includes the collection and analysis of environmental samples from 36 patient rooms at 3 major hospitals located in 3 different US states. The findings suggest that while complete eradication of pathogens to undetectable levels, which is the goal of most terminal cleaning protocols, was not consistently achieved, concentration levels of Acinetobacter baumannii, MRSA, VRE, and C diff. were moderately to significantly reduced after the cleaning procedure across different surface types. One interesting finding was the increase in VRE concentration on nurse call buttons after cleaning in one of the hospitals, which the authors argued to be potentially due to cross contamination. Another interesting finding, which agrees with other studies, was that using only ATP measurement was shown to be insufficient for assessing hospital surface cleanliness. The authors have provided suggestions about amending and standardizing current terminal cleaning protocols to protect patients from exposure to residual pathogens on surfaces left by previous patients, with a particular focus on MRSA and VRE as the most frequently found pathogens on hospital surfaces after terminal cleaning. In light of uncertainties regarding the effectiveness of terminal cleaning, the study suggests transitioning to a risk-based framework that can link terminal cleaning protocols to expected HAI outcomes. I found the study interesting and suitable for publication in PLOS ONE, though I have a few questions that I would like to be addressed before publication.

Major comments/questions

1. The authors have provided a brief discussion of the potential impacts of the differences in cleaning protocols and compliance across the three sites on their findings, in addition to other limitation of their study (which is shared with other similar studies). I think it would help to make a more comprehensive discussion if other factors that may contribute to variabilities and rare but significant discrepancies (such as VRE increase in one of the hospitals) were also incorporated, factors such as differences in cleaning staff diligence, patient colonization rates, admission prevalence rate, average length of stay, frequency of HCW visits, and other factors that affect pathogen shedding and transfer and may differ across rooms, wards, and hospitals.

a. This is a good point. We added this and additional factors know to contribute to microorganism transfer which should be examined in future studies.

b. Overall, microorganism detections and their concentrations decreased by surface from pre- to post-cleaning. During one sampling event in one room at hospital 1, the confirmed concentration of VRE increased from pre-cleaning (below limit of detection) to post-cleaning (0.15 CFU cm-2) sampling. This suggests cross contamination in cleaning hospital rooms is possible and requires further research on the spread of HAIs between fomites during cleaning practices. Surface contamination and transfer of microorganisms between fomites is complicated. While no information was gathered on patient behaviors, EVS cleaning practices and behaviors, frequency and duration of environmental cleaning, frequency of healthcare worker visit to patient room, airflow within hospital rooms, overall patient colonization rates in each hospital, infection prevalence rates at time of admission in each hospital, or average length of stay, these factors warrant future assessment as they could contribute to the transfer of pathogens between fomites. Future studies should also examine the role of cloth material and detergent wipes [31] and the surface roughness [32,33] in this transfer continuum. Increases in concentration following terminal cleaning could also indicate disaggregation during cleaning, increasing detectable CFUs. Given each of the three hospital cleaning protocols call for germicidal agents, the environmental sampling suggests additional cleaning efforts or prescribed surface cleaning orders are required to reduce pathogen occurrence and/or cross-contamination.

2. Discussion, Paragraph 2: “on at least one occasion, detections and/or concentrations increased from pre-cleaning to post-cleaning sampling. This occurred for C. difficile on chair handrail, MRSA on bed handrail…” While the “one” occasion was covered in the results section for VRE on nurse call buttons, other instances of such observations, where pathogen concentration was increased after terminal cleaning, was not explained neither could be interpreted from the results (Tables 2 and 3), for example C. difficile on chair handrail. I am not sure if these other instances were isolated cases in one or just a few rooms, or if the findings were not reliable due to below-detectable concentrations. What could be the reason? Please elaborate.

a. This paragraph was modified to focus only on the confirmed measurements to improve clarity and importance of surface testing methods. 

b. Overall, microorganism detections and their concentrations decreased by surface from pre- to post-cleaning. During one sampling event in one room at hospital 1, the confirmed concentration of VRE increased from pre-cleaning (below limit of detection) to post-cleaning (0.15 CFU cm-2) sampling. This suggests cross contamination in cleaning hospital rooms is possible and requires further research on the spread of HAIs between fomites during cleaning practices. Surface contamination and transfer of microorganisms between fomites is complicated. While no information was gathered on patient behaviors, EVS cleaning practices and behaviors, frequency and duration of environmental cleaning, frequency of healthcare worker visit to patient room, airflow within hospital rooms, overall patient colonization rates in each hospital, infection prevalence rates at time of admission in each hospital, or average length of stay, these factors warrant future assessment as they could contribute to the transfer of pathogens between fomites. Future studies should also examine the role of cloth material and detergent wipes [31] and the surface roughness [32,33] in this transfer continuum. Increases in concentration following terminal cleaning could also indicate disaggregation during cleaning, increasing detectable CFUs. Given each of the three hospital cleaning protocols call for germicidal agents, the environmental sampling suggests additional cleaning efforts or prescribed surface cleaning orders are required to reduce pathogen occurrence and/or cross-contamination.

3. The one occasion where VRE concentration was increased after terminal cleaning is interesting. While the authors have provided their interpretation of the potential causes, it would make it easier for readers to objectively interpret such a rare but impactful case if more information was provided, for example, was this observed for all nurse call buttons (all rooms), if not how many of the rooms, and whether there could be anything different about those specific rooms or their occupying patients. This deserves a little deeper investigation.

a. This was added to the paragraph. The paragraph is provided in the response to the previous comment. 

4. Statistical analysis: what method was used for multiple hypothesis tests (detections) to correct for family-wise type I errors? If such a consideration was not necessary, please justify by explaining in the paper the scale of detection tests (i.e., whether detection tests were done on a surface-, room-, or hospital-level) in relation to the comparison scale which is surface type at each hospital.

a. Thank you for catching this. Due to the small, confirmed sample sizes on each surface and hospital, comparative analysis and ad hoc testing was not performed. This method text was removed from the paper and replaced with a clarification sentence. 

Minor comments

1. Abstract: Please spell out what ATP stands for, the first time it is used.

a. Completed

2. Abstract: Incomplete sentence: “Surfaces from 36 occupied … and Clostridioides difficile (C. difficile)).”

a. Completed sentence

b. Surfaces were swabbed from 36 occupied patient rooms with a laboratory-confirmed, hospital- or community-acquired infection of at least one of the four pathogens of interest (i.e., Acinetobacter baumannii (A. baumannii), methicillin resistant Staphylococcus aureus (MRSA), vancomycin resistant Enterococcus faecalis/faecium (VRE), and Clostridioides difficile (C. difficile)).

3. Abstract: Missing comma before “desk surface” in the following statement: “Six nonporous, high touch surfaces (i.e., chair handrail, bed handrail, nurse call button desk surface, bathroom counter near the sink, and a grab bar near the toilet) ...”

a. Added comma

4. Introduction, Paragraph 1: “ATP as a biochemical indicator of biological material.” Incomplete sentence. Please complete or merge with the previous or next sentence.

a. Modified sentence

b. ATP is a biochemical indicator of biological material.

5. Methods, Sample Site Selection, Paragraph 2: Missing comma before “desk surface” in the following statement: “The six surfaces (i.e., chair handrail, bed handrail, nurse call button desk surface, bathroom counter near the sink, and a grab bar near the toilet) ...”

a. Added comma

6. Methods, Sample Site Selection, Paragraph 2: Please address the cross-referencing error.

a. This has been corrected to “Table 1”

7. Methods, Sample Site Selection, Paragraph 2: “If a pre-selected item was not present …” An example or two for the replacements could help clarify, also how often this was the case?

a. This sentence was left over from our field sampling protocol. The surfaces selected were found in all rooms sampled during this project. The sentence has been removed. 

8. Methods, Sample Site Selection, Table 1: To improve readability, please align cells, preferably to the top left corner of the cells, or use borders.

a. This was completed but believe the final format will be addressed by PLOS production team

9. Methods, Detections, below Figure 1: “At hospital 2, VRE was the only organism detected and confirmed (n nurse call button) pre-cleaning…” Please correct the typo inside the parentheses.

a. Deleted

10. Discussion, Paragraph 4: “Other studies, however, have shown poor relationship relationships [38–41] …” Please remove the extra “relationship.”

a. Removed

Reviewer #2: Thank you for this informative paper in this important area of infection prevention and control in the hospital setting. I have some comments that revolve primarily around making the paper clearer, and easier for the reader to understand and contextualize. It includes using a denominator to give context to the numbers you are referrring to.

2. The abstract needs to include the number of rooms being evaluated for each in order to make sense of these results.

a. The number of rooms was included now for detections listed in the abstract. 

3. The statement "ATP detections did not correlate with any pathogen concentration and thus are not recommended for assessing cleanliness or safety." is too conclusive based on the data shown. ATP measures organic matter, and would be better correlated with overall aerobic colony counts, it is not specific to multidrug resistant organisms.

a. The sentence was modified to focus on our findings only and does not include any recommendations

b. ATP detections did not correlate with any pathogen concentration.

4. In methods, there is an error: "Descriptions of the surfaces are presented in Error!Reference source not found.."

a. This has been corrected to “Table 1”

5. Include use of negative control in the micro methods section if used, or if not used explain why.

a. We thank the reviewer for the critical methods assessment. We did complete negative controls during this project and we have included that information in the methods now. 

a. Added: A negative control was performed for each sampling event. This was completed by opening a swab, breaking the sponge end off into the bag, and leaving the sponge in the bag during room sampling. The bag was then closed and processed in the laboratory alongside the surface samples.

6. Presumptive and confirmed concentrations needs to be explained in the methods. From my reading/understanding only AB isolates from wet surfaces were confirmed as these may be non-baumanii. But then it seems to apply to all and is very unclear. Table 2 is also very unclear.

a. This is helpful feedback. To be clear, confirmation was completed for C. difficile, MRSA, or VRE but not A. baumannii. The authors have added text and modified language to improve method clarity. 

b. This sentence was added to the beginning of the second paragraph in ‘microbial analysis’ section: Confirmation testing was completed for C. difficile, MRSA, or VRE.

c. This sentence was added in the middle of the same paragraph: Confirmation testing was not completed for Acinetobacter spp.

d. Description of concentration calculation methods were also added: A sample was considered a confirmed detection when 1) a sample had at least one presumptive CFU on selective agar and 2) at least one presumptive CFU was confirmed as the target pathogen through MALDI-TOF. The final confirmed concentration for each organism was calculated using the presumptive concentration multiplied by the fraction of isolates confirmed positive through MALDI-TOF assessment with results presented as CFU/swabbed area.

e. Table 2 format has been adjusted and a footnote has been included: Less than values indicate concentrations below the limit of detection.

7. Detections needs a denominator to put the “51” and “12” on the first line in context. The rest of the section is unclear, and does not seem to align with the methods.

a. These denominators have been added in the sentence. 

b. The paragraph has been modified to improve clarity. 

8. Discussion: needs to be clearer - Other studies, however, have shown poor relationship relationships [38–41] or significant relationships [42–47] between ATP and pathogen presence.

a. The language in this paragraph has been modified to address this comment and improve discussion clarity. 

b. ATP results from the current study were not a reliable indicator of pathogen presence or cleaning efficacy. Across all hospitals, ATPs were measured on nearly all surfaces before and after cleaning. This supports prior hospital studies in the USA [35,36], and Brazil [37] that found no significant relationship of ATP and pathogens. While numerous studies have explored the association between ATP levels and the presence of pathogens in hospital settings, the existing body of literature reveals a spectrum of findings. Some studies report poor relationships [38–41], while others indicate significant associations [42–47] between ATP measurements and pathogen presence. This variability underscores the complexity of assessing cleaning practices and the importance of utilizing multi assessment approaches to understand environmental hygiene. This approach is crucial for refining and enhancing infection control strategies, ultimately contributing to the overall improvement of patient safety in healthcare facilities.

---

## [Decision Letter · Decision Letter 1]

24 May 2024

Impact of terminal cleaning in rooms previously occupied by patients with healthcare-associated infections

PONE-D-23-27473R1

Dear Dr. Verhougstraete,

We’re pleased to inform you that your manuscript has been judged scientifically suitable for publication and will be formally accepted for publication once it meets all outstanding technical requirements.

Kind regards,

Eili Y. Klein, PhD

Academic Editor

PLOS ONE

Additional Editor Comments (optional):

Thank you for your diligence in responding to all the comments and your patience in the process

Reviewers' comments:

Reviewer's Responses to Questions

**Comments to the Author**

1. If the authors have adequately addressed your comments raised in a previous round of review and you feel that this manuscript is now acceptable for publication, you may indicate that here to bypass the “Comments to the Author” section, enter your conflict of interest statement in the “Confidential to Editor” section, and submit your "Accept" recommendation.

Reviewer #1: All comments have been addressed

Reviewer #2: All comments have been addressed

2. Is the manuscript technically sound, and do the data support the conclusions?

Reviewer #1: Yes

Reviewer #2: Yes

3. Has the statistical analysis been performed appropriately and rigorously? 

Reviewer #1: Yes

Reviewer #2: Yes

4. Have the authors made all data underlying the findings in their manuscript fully available?

Reviewer #1: Yes

Reviewer #2: Yes

5. Is the manuscript presented in an intelligible fashion and written in standard English?

Reviewer #1: Yes

Reviewer #2: Yes

6. Review Comments to the Author

Reviewer #1: (No Response)

Reviewer #2: Thank you for addressing the reviewer comments. I do not have any further comments or edits to suggest.

7. PLOS authors have the option to publish the peer review history of their article (what does this mean?). If published, this will include your full peer review and any attached files.

Reviewer #1: **Yes: **Fardad Haghpanah

Reviewer #2: No

---

## [Editor Report · Acceptance letter]

1 Jul 2024

PONE-D-23-27473R1 

PLOS ONE

Dear Dr. Verhougstraete, 

I'm pleased to inform you that your manuscript has been deemed suitable for publication in PLOS ONE. Congratulations! Your manuscript is now being handed over to our production team.

Kind regards, 

on behalf of

Dr. Eili Y. Klein 

Academic Editor

PLOS ONE